# An Overview of Emergent Order in Far-from-Equilibrium Driven Systems: From Kuramoto Oscillators to Rayleigh–Bénard Convection

**DOI:** 10.3390/e22050561

**Published:** 2020-05-17

**Authors:** Atanu Chatterjee, Nicholas Mears, Yash Yadati, Germano S. Iannacchione

**Affiliations:** Department of Physics, Worcester Polytechnic Institute, 100 Institute Road, Worcester, MA 01605, USA; nemears@wpi.edu (N.M.); yyadati@wpi.edu (Y.Y.)

**Keywords:** non-equilibrium thermodynamics, Ising model, Kuramoto model, Rayleigh–Bénard convection, pattern formation

## Abstract

Soft-matter systems when driven out of equilibrium often give rise to structures that usually lie in between the macroscopic scale of the material and microscopic scale of its constituents. In this paper we review three such systems, the two-dimensional square-lattice Ising model, the Kuramoto model and the Rayleigh–Bénard convection system which when driven out of equilibrium give rise to emergent spatio-temporal order through self-organization. A common feature of these systems is that the entities that self-organize are coupled to one another in some way, either through local interactions or through a continuous media. Therefore, the general nature of non-equilibrium fluctuations of the intrinsic variables in these systems are found to follow similar trends as order emerges. Through this paper, we attempt to find connections between these systems, and systems in general which give rise to emergent order when driven out of equilibrium. This study, thus acts as a foundation for modeling a complex system as a two-state system, where the states: order and disorder can coexist as the system is driven away from equilibrium.

## 1. Introduction

A system at equilibrium is indistinguishable from its surroundings. The same system when driven out of equilibrium gives rise to flows that force the system to relax back into its equilibrium state. The rate of relaxation is governed by how far the system has been driven out of equilibrium [1,2,3]. Soft-matter systems in this respect are especially fascinating as they often give rise to order as long as the driving field maintains it out of equilibrium [1,4,5]. Some prominent examples where self-organization gives rise to emergent order include, liquid crystals, granular material, polymers, gels, networks, and a wide spectrum of biological phenomena/materials [6,7,8,9,10,11,12,13,14,15,16,17]. This emergent order can vary across several lengths and time scales, and since they are very sensitive to fluctuations (thermal) they are usually difficult to predict.

In this paper, we discuss how coupling plays an important role in driven systems as they self-organize to give rise to emergent order. In driven systems, the emergence of order is mediated by the presence of numerous irreversible processes within the system through a continuous exchange of energy between the system and the surrounding. As noted by Demirel, these interactions are called thermodynamic couplings, which may allow a process to progress without its primary driving force or in a direction opposite to the one imposed by its own driving force [18]. We start with the simplest statistical model that shows phase transition—the two-dimensional square lattice Ising model. Following which, we model the phenomena of synchronization of a large set of coupled oscillators or the Kuramoto model. Both of these systems are numerically modelled and the effect of mean-field coupling on the emergence of order is discussed. The results are then compared with an experimental system, the Rayleigh–Bénard convection for different fluid mixtures at varying Rayleigh numbers ($10^{3}$–$10^{6}$). In all of these systems, the emergence of order is mediated by various irreversible processes inside the system which results in a process of continuous exchange of energy with the surrounding media. The onset of order in these systems is controlled by the critical value of the coupling strength, for example, a critical temperature in the case of a two-dimensional Ising model exhibiting phase transition, critical coupling strength in a collection of Kuramoto oscillators, and a critical Rayleigh number in a Rayleigh–Bénard instability. Since the focus of this study lies in the search for simple modeling tools to understand pattern formation in a complex system, the choice of our model systems is not random. In our previous studies, we had shown how the mean temperature of the top layer of the fluid film bifurcates into ‘hot’ and ‘cold’ domains as a steady-steady is achieved in a non-turbulent Rayleigh–Bénard system [19,20,21]. The coexistence of two states, hot and cold, draws similarities to the Ising model which has been successfully used to understand many two-state systems/processes. Along similar lines, coexistence of synchronous and asynchronous oscillators in Kuramoto systems have given rise to studies on chimera states [22]. Therefore, we use these model systems as test subjects to understand the nature of emergent order and its connection to the second statistical moment of the respective intrinsic variables by comparing their similarities and differences across them. For instance, it is observed as a general result that microscopic fluctuations tend to decay as macroscopic order emerges in these systems, however, the nature of the decay varies considerably and is often dictated by how ‘far’ they have been driven out of equilibrium. In conclusion, this study acts as a foundation for modeling a complex system as a two state-system, where the states: order and disorder coexist as the system is driven away from equilibrium.

### 1.1. Ising Model

One of the earliest physical models that studied the emergence of order as a consequence of interaction between agents is the Ising model [23,24]. Traditionally, the Ising system was used to model ferromagnetism in statistical physics, where magnetic dipoles could either have a spin ‘up’ or ‘down’. Since then, it has become a prototype for many two-state model system examples including, protein folding, ligand–receptor interactions, spin glasses, firing of neurons etc. [25,26,27,28]. The Hamiltonian for an Ising system in the presence of an external field ‘*h*’ is given by,
(1)$$H=\sum_{ij}J_{ij}\sigma_{i}\sigma_{j}-\sum_{j}h_{j}\sigma_{j}$$

The spins are denoted by σ and the indices represent neighboring lattice sites. The signature of $J_{ij}$ tells us the nature of interaction between the pair (i,j). While the simplest case of the Ising system is the one-dimensional case, interesting features emerge when it is studied on a two-dimensional square lattice. The two-dimensional square-lattice Ising model is one of the simplest statistical models that allows for phase transition [23,24,29]. In order to numerically solve the problem, a two-dimensional partition function is defined,
(2)$$Z\left(m,n\right)=\sum_{\sigma }\exp\left(m\sum_{i,j}\sigma_{i}\sigma_{j}+n\sum_{i,j}\sigma_{i}\sigma_{j}\right)$$

Here, σ assigns a value of either +1 (up) or −1 (down) for each lattice site and the variables ‘*m*’ and ‘*n*’ denote the rows and columns of the lattice (the special case being m=n=N) with periodic boundary conditions. For the case of isotropic coupling one achieves a phase transition when,
(3)$$\beta_{c}=\frac{\ln(1+\sqrt{2})}{2}\approx 0.4,\;\beta_{c}=1/k_{B}T_{c}$$

In the above equation, β represents the inverse temperature, $k_{B}$ the Boltzmann’s constant, and $T_{c}$ the Curie temperature. For large systems it is impossible to calculate statistical averages directly. The dynamics of the model was therefore simulated using a Monte Carlo algorithm to approximate real thermal averages while randomly assigning spin values at every lattice site. One can encounter a practical problem if the spins are to randomly flip at every lattice site with each simulation step, and end up eventually with a checkerboard pattern. Therefore, random walks are used to take into account only important spins configurations by introducing a Markovian decision model where the spin state at a site is the most probable outcome based on spin probabilities in a set of randomly chosen sites within the lattice. The transition probability from one configuration state to another is determined by the energy of the two configuration states. If ΔH>0, then the transition probability takes the form, exp(−ΔH/T); however, if ΔH<0 the transition probability is 1 as the system transitions to a state of lower energy. While better prediction based on larger regions for decision making makes the simulation faster, this was not really the aim of the model. Macroscopically, the system’s dynamics is ‘equilibrium-like’, but microscopically spin outcomes at each lattice site is inherently stochastic. The emergence of order was further analyzed when the system was externally perturbed under conditions: h=constant and h(t)=Asinωt (refer Equation (Equation 1)). More details on the simulation and the codes can be found here (see supplementary files) [30].

Although the Ising model has played a central role in the study of equilibrium phase transition, it is important to note that it provides insights which are of a general nature. The decrease in the standard deviation of the magnetization as order emerges in an Ising model forms the basis of the current study as it connects other driven systems where pattern emerges as fluctuations in the system decay, for example, the coupled Kuramoto oscillators and the Rayleigh–Bénard system.

### 1.2. Kuramoto Model

Similar to the Ising system, one can model collective synchronization in a large population of oscillating elements. The Kuramoto model is a mathematical model that treats a system as an ensemble of limit-cycle oscillators described only by their phases [31,32,33]. In the simplest version of the model, each oscillator in the Kuramoto system has its own intrinsic natural frequency $\omega_{i}$ and is coupled to every other oscillator in the system. The intrinsic natural frequencies of the oscillators are drawn from a predefined distribution, usually a normal distribution with well-defined mean and standard deviation. As the system collectively synchronizes, the different frequencies spontaneously locks to a common frequency, Ω. The Kuramoto model has found several successful applications in condensed matter physics specially in the study of biological phenomena and active matter [32,34,35]. The governing equation for the system is given by,
(4)$$\frac{d\theta_{i}}{dt}=\omega_{i}+\frac{\kappa }{N}\sum_{j=1}^{N-1}\sin\left(\theta_{j}-\theta_{i}\right),\;j\neq i$$

Here, the phase of an oscillator is given by $\theta_{i}$ and the coupling strength by κ. Through the following transformation one can solve this nonlinear differential equation for the mean-field case, N→∞:(5)$$Re^{i\phi }=\frac{1}{N}\sum_{j=1}^{N}e^{i\theta_{j}}$$
where *R* is the order parameter and ϕ the average phase, one can transform Equation (Equation 4) and rewrite it as,
(6)$$\frac{d\theta_{i}}{dt}=\omega_{i}+\kappa R\sin\left(\phi -\theta_{i}\right)$$

Since the oscillators are randomly oriented, the sum over all oscillator phases average to zero. Hence, Equation (Equation 6) becomes,
(7)$$\frac{d\theta_{i}}{dt}=\omega_{i}-\kappa R\sin\left(\theta_{i}\right)$$

For sufficiently strong coupling one achieves a fully synchronized state (R→1). At a fully synchronized state all the oscillators share a common frequency while their phases may differ. Under steady-state condition ($d\theta_{i}/dt=0$), the fully synchronized solution for Equation (Equation 7) reduces to $\omega_{i}\to \Omega =\kappa \sin\left(\theta_{i}\right)$ where Ω is the common frequency of the oscillators. In this study, the mean-field Kuramoto model was simulated on a two-dimensional square lattice. The effect of coupling strength was observed on the time evolution of the order parameter and simultaneously on the second statistical moment of the angular frequencies of the oscillators. The effect of several other types of coupling mechanisms were also studied (like distance-dependent inverse square), but are not presented in this paper. In this context, interested readers are referred to [30].

### 1.3. Rayleigh–Bénard Convection

Finally, we discuss one of the simplest experimental setups to study pattern formation and self-organization. As a thin layer of viscous fluid is heated and convection sets in, one can observe thermal gradients on the surface of the fluid film which are stable in time. The regular pattern of convection cells are known as Bénard cells and the phenomena, Rayleigh–Bénard convection [4,36,37,38]. To date, it remains one of the most actively and extensively studied physical systems. Due to its conceptual richness, the dynamics of a Rayleigh–Bénard convection phenomena connects fundamental ideas from both non-equilibrium thermodynamics and fluid mechanics [39,40,41]. The beauty of this system lies in its simplicity, wherein the critical value of a dimensionless quantity, the Rayleigh number (Ra), determines the onset of convection. The Rayleigh number relates the physical quantities, *g* (acceleration due to gravity), β (thermal expansion coefficient), ΔT (temperature difference across the fluid film thickness), *l* (fluid film thickness), ν (kinematic viscosity) and α (thermal diffusivity) as below,
(8)$$Ra=\frac{g\beta \Delta Tl^{3}}{\nu \alpha }$$

The critical Rayleigh number, $Ra_{c}$ of 1708 marks the onset of convection for a no-slip boundary condition was obtained by Jeffreys in 1929 [4,38]. In our study, a simple setup for the Rayleigh–Bénard consists of a top cover and a bottom base on which a copper pan is placed. The top cover is made up of wood and has ducts for forced convective heat transfer. The thermocouples attached to the ducts measure the temperature of the incoming and outgoing gas. The bottom rest, also made up of wood has a cavity with a recess on which the copper pan sits snugly. The wooden base rests on top of a block of foam. A thermocouple and a heater is attached to the base of the copper pan which measures the bottom temperature of the pan. An infra-red camera is placed at a height above the copper pan which captures the real-time thermal images of the convection cells. The temperature scale of the camera is calibrated by heating the empty copper pan. The data obtained in this study are grey-scale thermal images that can be converted into a temperature matrix. The temperature of the top layer of the fluid film is obtained by averaging over the entire exposed area. With this setup in place, two types of studies are performed: spatial and temporal. In the temporal study, thermal statistics are recorded from a room temperature equilibrium to a non-equilibrium steady-state as the system is thermally driven by regulating the power input through the heater. Whereas, in the spatial study thermal statistics are obtained once the system has reached a non-equilibrium steady-state. While the temporal study allows us to envision the evolution of order in the system, the spatial study lets us visualize how order is spatially distributed through emergent length-scales. One can find more details on the study: the experiments and the analyses here [20,21].

## 2. Results

In this section, we compare the results from our numerical simulations and the experimental study for the three systems. We are essentially looking at the possibility that these systems which are distinctly different from one another exhibit characteristics that are similar when driven out of equilibrium. It will also be interesting to discuss how they differ as well. In Figure 1, we report our results from the simulation of the two-dimensional square lattice Ising model. Figure 1a plots magnetization as a function of inverse temperature ($\beta =1/k_{B}T$). The magnetization ($S/\bar{S}$) acts as the order parameter for the system. The ferromagnetic transition happens around β≈0.44 which corresponds to the Curie temperature (see Equation (Equation 3)) [24]. The Ising system is then perturbed by an external field, and the evolution of the order parameter is plotted as a function of time in Figure 1b and as a function of inverse temperature in Figure 1c. In the case of no external field one can see that the spins eventually lock into a mixed state with some of the lattice sites with, say ‘up’ spin and the remaining with ‘down’ spin. Therefore, the system does not reach a fully spin ‘up’ or spin ‘down’ state as seen from Figure 1b. This, however, is not the case when there is an external perturbation as the system eventually directs itself to the direction of the external perturbation as that is energetically more favorable. In case of a sinusoidal time varying field, some oscillations are observed because of periodic aligning and re-aligning. The smaller the temperature, the larger the β and as theory predicts we see phase transitions at sufficiently low enough temperature in Figure 1c. Therefore, for no external perturbation, the transition temperature seems to be the lowest at β≈0.4. In Figure 1d, we plot the standard deviation of the spin as a function of time for the case of no external perturbation at β=0.2. It is observed that the standard deviation, being a measure of fluctuation, steadily decreases as order emerges in the system.

In Figure 2, we plot the scaled standard deviation of the intrinsic variable and emergent order as a function of time for the Kuramoto system and the Rayleigh–Bénard convection. The intrinsic variable in the Kuramoto system is the angular frequency of the oscillators ($\omega_{i}$) which collapses to a common frequency (Ω) as the system achieves synchronization. In the Rayleigh–Bénard system spatially-averaged temperature (〈T(t)〉) plays the same role. As the system approaches a steady-state, $\langle T\left(t\right)\rangle \to \langle T_{\infty }\rangle$, where $\langle T_{\infty }\rangle$ is the spatially-averaged steady-state temperature of the system. In Figure 2a, we plot the scaled standard deviation for the Kuramoto model as a function of time for two values of the coupling strength, κ=1.5 and κ=2. It is evident from the theory and the plot in Figure 2e that order (*R*, defined in Equation (Equation 5)) emerges faster in the case of higher coupling strength. At time-step, t=100 one can observe that atleast more than half of the oscillators present in the system are synchronized (from Figure 2e) and therefore one observes a sharp decline in the scaled standard deviation plot in Figure 2a. Later one can notice that as t≥110 there is a sudden spike in the standard deviation as order increases further. The reason for this could be attributed to a mixture of synchronized and unsynchronized oscillators as R<1. As time progresses, the natural frequencies of all the oscillators approach closer to mean-field common frequency. However, due to their equally random phase orientations, some of the oscillators reach the common frequency and lock themselves in that state earlier than the other. A situation like this although reduces the standard deviation when compared to the randomized initial state it however increases the standard deviation at an instant when these two groups of oscillators start oscillating simultaneously, one with low fluctuations and the other with higher fluctuations. As one would expect, this scenario appears to last longer in the case of lower coupling strength among the oscillators because of more unsynchronized oscillators than synchronized ones at any given instant in time.

Following our results from the Kuramoto system, we look at the Rayleigh–Bénard convection in the remaining panels of Figure 2. In our experiments we use three fluid samples: silicone oil, glycerol and glycerol-water mixture (1:4 and 1:2 by volume). The three fluid samples allow us to explore a wide range of Rayleigh numbers. There is no well-defined order parameter in a Rayleigh–Bénard system, therefore we define one based on the thermal profile at steady-state as,
(9)$$R=\frac{\langle T\left(t\right)\rangle -\langle T_{0}\rangle }{\langle T_{\infty }\rangle -\langle T_{0}\rangle },\;\mathrm{such}\;\mathrm{that}\;0\leq R\leq 1,\;\mathrm{when}\;\langle T_{0}\rangle \leq \langle T\left(t\right)\rangle \leq \langle T_{\infty }\rangle$$

Here, 〈T(t)〉 represents spatially-averaged temperature at any instant in time, $\langle T_{0}\rangle$ represents spatially-averaged temperature at initial equilibrium state (room temperature) and $\langle T_{\infty }\rangle$ represents spatially-averaged temperature at a non-equilibrium steady-state. For each of the fluid samples, we look at the scaled standard deviation plots as order emerges. In Figure 2b, we plot the results from our studies on the silicone oil sample with viscosity ν=150 cSt, and low Rayleigh numbers for $\Delta T\approx 30^{{}^{\circ}}$–$60{\;}^{{}^{\circ}}$C. From the figure, we can observe that the thermal fluctuations increase gradually upto t∼100 s, and then gradually decrease from t∼10–$10^{3}$ s. Following which, they keep increasing till the system reaches a steady-state. We have discussed this aspect of decline in thermal fluctuations at the onset of emergent patterns in our previous studies on the Rayleigh–Bénard system [20,21]. It was found, that these fluctuations reach a minima when a certain number of cells start to nucleate at the center of the copper pan. The decline continues until a stable emergent pattern, consisting of convection cells and rolls covers the entire top layer of the fluid film. As the fluid sample is highly viscous, these convection cells are stable in time and space in the non-turbulent regime. Therefore, they do not nucleate or divide any further. From Figure 2f, we can see that the system takes another ∼$10^{3}$ time-steps to reach a steady-state, thus the bulk temperature and fluctuations due to thermal agitation continue to rise further. With Rayleigh numbers in the similar range, we however see a very different characteristic when glycerol is our working fluid. As glycerol is a fluid whose viscosity is one-tenth that of silicone oil, we observe that the nucleation and subsequent emergence of the first set of convection cells is immediately followed by rapid division into smaller cells. This two-step process of division results into two declining trends in the standard deviation plot as shown in Figure 2c, first between t∼ 300–700 s and the second between t∼ 2000–8000 s. The magnitude of the fluctuation is visibly higher during the second decay as the system is still approaching a state-steady which is inline with our observations on the silicone oil sample. In Figure 2d, we study the thermal fluctuations in glycerol-water mixtures. It is observed that the decline in the standard deviation is much more rapid as compared to the earlier cases. Quite interestingly, the onset of convection with the appearance of the first few cells and the subsequent spread throughout the top layer of the fluid film happens within a span of 100 s, between t∼ 100–200 s. The reason for this appears to be very low viscosities (∼$10^{-2}$ cSt) which yields very higher Rayleigh numbers. Therefore, nucleation not only happens early but also spreads at a faster rate throughout the pan. Following which, they break down into smaller and smaller domains and start dissipating heat chaotically as the system enters a turbulent regime. One can observe this from the amount of noise in the standard deviation plots (almost immediately after t∼200 s). As the system asymptotically reaches a steady-state (see Figure 2h), the magnitude of this thermal noise due to chaotic thermal agitation keeps growing with time.

One quick look at the plots is enough to highlight the differences between the three convection systems and the Kuramoto model. However, what is striking is the presence of one common feature in the standard deviation plots across all the systems. The observation that there is a decline in the standard deviation is the common feature that connects them. The dissimilarities such as, the presence of multiple peaks and troughs, different rates of nucleation/synchronization etc. originate from the fact that there exist multiple time-scales in these systems. There is one global time-scale that dictates the bulk equilibration (steady-state) or global synchrony, and there are multiple local time-scales such as, individual frequencies of the oscillators or the time taken by convective cells/rolls to locally equilibrate. Neither are these time-scales comparable across systems, nor do they overlap within a particular system. Therefore, these are systems at steady-states that have been locally equilibrated far away from equilibrium. The emergence of spatial ordering of convection cells in the case of non-turbulent Rayleigh–Bénard system implies the presence of an emergent work that drives a volume of fluid from the bottom of the pan to the top while dissipating heat laterally. These stable spatial structures are a set of locally equilibrated convection cells that are synchronous with a common frequency, $\omega =d\theta /dt=2u_{\infty }/l$ where l/2 is the half thickness of the fluid film and the steady-state velocity, ${\vec{u}}_{\infty }\propto \nabla T$ where ∇T is the thermal gradient across the fluid film thickness.

In Figure 3, we plot the probability densities of the scaled fluctuation of the intensive variables for the initial randomized state and compare them with the final synchronized state for the Kuramoto model and the Rayleigh–Bénard system. Fluctuation in the Kuramoto system is measured by the deviation of the natural frequency of an oscillator from the mean frequency of the system, δω=ω(t)−〈ω〉. This deviation in the natural frequencies of the oscillator is scaled by the mean frequency of the system, which we define as scaled fluctuation for the Kuramoto system, $\delta \omega^{⋆}=\delta \omega /\langle \omega \rangle$. Once the oscillators are fully synchronized, 〈ω〉→Ω. Similarly, in the Rayleigh–Bénard convection we define thermal fluctuation as δT=T(t)−〈T〉, and $\delta T^{⋆}=\delta T/\langle T\rangle$. At room-temperature equilibrium, $\langle T\rangle \to T_{0}$ and at steady-state, $\langle T\rangle \to T_{\infty }$. As an equilibrium state corresponds to symmetry conservation, one expects to obtain normal fluctuations in the initial state. In Figure 3a,b, we plot the scaled fluctuation distribution for the Kuramoto oscillators and the Rayleigh–Bénard convection respectively in their initial state. We can clearly see that the data obeys very well with the Gaussian fits centered around the origin. For the final fully synchronized state of the oscillators one would expect that a probability density function which would take the form of a delta function sharply centered at the origin such that,
(10)$$\delta \left(x\right)=\left\{\begin{array}{cc} 0 & x\neq 0\\ \inf & x=0 \end{array}\right.\;\mathrm{and}\;\int_{-\epsilon }^{+\epsilon }dx\delta \left(x\right)=1\;\mathrm{if}\;0\in \left[-\epsilon ,+\epsilon \right]$$

Note that in the above equation, $x=\delta \omega^{⋆}$. A realistic approximation to such a distribution when there are tails in the data is a Lorentzian function,
(11)$$\delta \left(x\right)=\lim_{\epsilon \to 0}\frac{1}{\pi }\frac{\epsilon }{x^{2}+\epsilon^{2}}$$

Therefore, a Lorentzian function of the form shown in Equation (Equation 11) fitted to the Kuramoto data for the final synchronized state is in good agreement as seen from Figure 3a. The tail present in the data is captured by the functional part which decays as, $1/x^{2}$ in the neighborhood of 0∈[−ϵ,+ϵ]. In the case of the Rayleigh–Bénard convection we cannot expect to see a single sharply peaked distribution centered around the origin for the scaled thermal fluctuations. As we can see from Figure 3c,d, the data shows the presence of two peaks (or bimodality). The bimodal distribution in the thermal fluctuations originates from the fact that there are upward and downward drafts as the fluid element completes a convection cycle between the bottom hot and the top cold surface. In Figure 3c, we plot the kernel density estimates to determine the shape of the probability density function for the two experimental trials with different Rayleigh numbers. In Figure 3d, we proceed to fit the data piece-wise. We choose individual tails and fit them with a pair of Gaussian fit functions (in black) and then with a pair of Lorentzian fit functions (in red). As we can see from our plots in Figure 3d, both Gaussian and Lorentzian fits superimpose over one another. The difference between the center of the two peaks is about 0.04 units with one peaking in the positive domain and the other in the negative. Therefore, one peak signifies the contribution of the upward plumes and the other of the the downward plumes. We are still unsure of the fact that how the fit functions from the two tails merge into one another. In some of our recent works we discuss the presence of a mixture of local equilibrium regions in the Rayleigh–Bénard convection which describes the bimodal nature of the thermal fluctuations [19,20,21]. To conclude, in the mean-field Kuramoto model the final synchronous state being unique allows for the existence of a sharply peaked delta-type distribution, which in reality is best illustrated by a Lorentizian fit. In the case of a non-turbulent Rayleigh–Bénard convection at steady-state we find that there exist two possible states due to the existence of spatial thermal gradients which are stable in time. These stable spatial gradients lead to the emergence of two local equilibrium-like regions, fluctuations within which can be best represented by respective Gaussian distributions [21].

In Figure 4, we plot the lattice entropy as a function of time for a two-dimensional Kuramoto model with high coupling strength. We have previously seen that evolution of order in the system is inversely related to the fluctuations of the intrinsic variables. By calculating the Shannon entropy summed over every lattice site at every instant in time, we look at the relationship between entropy reduction and fluctuation decay as order emerges in the system. The notation, $\omega_{ij}$ denotes the frequency of the *i*th oscillator at the *j*th lattice site.
(12)$$S\left(\rho \right)=-\sum_{i}\sum_{j}\rho \left(\omega_{ij}\right)\ln\rho \left(\omega_{ij}\right)$$

Therefore, to make sure that the Kuramoto system reaches a fully synchronized state, a high enough coupling between the oscillators is chosen and the simulation is run for a very long duration. It is not surprising that at κ=5, the model achieves complete synchronization after just 500 time-steps. The Shannon entropy is calculated from Equation (Equation 12) and is scaled by its maximum value, such that $0<S/\bar{S}\leq 1$ [27,42]. As order emerges, one can clearly see from Figure 4 that a reduction in system’s entropy is accompanied by a reduction in the fluctuation of the intrinsic variable.

## 3. Conclusions

In this paper, we consider three systems that can be externally perturbed and driven out of equilibrium. While the Ising model and the Kuramoto oscillators are numerically solved, the Rayleigh–Bénard convection on the other hand was experimentally probed. The common feature of all the three systems is the emergence of order as they are driven out of equilibrium. The Ising and the Kuramoto models self-organize by ordering their spins and synchronizing their natural frequencies. On the other hand, the spatio-temporal order that emerges in the case of a Rayleigh–Bénard convection is a result of the competing forces between viscosity and buoyancy which gives rise to convective instabilities. Conceptually, the Rayleigh–Bénard system is richer and much more difficult to model than the other systems that were studied. However, it holds a prominent place in the study of complex systems, and in general systems that show pattern formation when driven out of equilibrium. Our goal of this study is to propose a general framework that can be used to connect these seemingly different systems by a common thread. An observation, that was found to be consistent across the three systems, was that the fluctuations of the intensive variables decay as order emerges. This observation is non-trivial because if a system is driven out of equilibrium, say thermally, the natural outcome is to expect the fluctuations to grow as a function of time due to thermal agitation/collision and momenta exchange. In this study, as well as in our previous works, we have shown that emergence of order is followed by a decline in fluctuation. In the light of our experimental results we believe this is an important result because we have obtained experimental evidence that shows how far-from-equilibrium fluctuations dictate pattern formation. When the Rayleigh–Bénard system is driven out of equilibrium but the Rayleigh number is below the critical value we see no emergent spatio-temporal patterns accompanied by a growth in the magnitude of the non-equilibrium fluctuations. However, when the Rayleigh number exceeds the critical value, we observe the emergence of macroscopic patterns and a simultaneous decrease in non-equilibrium fluctuation [21].

It is also important to note that the Ising model has been very successful in describing many two-state systems. The fact that multiple states can coexist (Rayleigh–Bénard: hot/cold and Kuramoto model: synchronous/asynchronous) when the system has been macroscopically driven out of equilibrium naturally allows us to frame the problem of pattern formation as a two-state problem. Infact, there are examples of oscillator based Ising machines and Ising models of turbulence in fluid [43,44]. Also, in our experimental studies we found that for high Rayleigh numbers, we enter the turbulent regime which gives rise to structures that are found to be unstable in time. We imagine that a system can only give rise to stable patterns when it is not driven too far away. This brings us to a more pertinent question as to how far away are these systems from equilibrium? Although we do not yet have a metric to define that but we can anticipate that these systems, even when driven out of equilibrium, are in a state of quasi-equilibrium where the local equilibrium hypothesis is satisfied [45,46]. Therefore, the end states are either stable equilibrium states or they are ‘equilibrium-like’ states which show equilibrium-like fluctuations. The sole focuses of this study are the end states: the room temperature equilibrium state and the out-of-equilibrium steady-state, therefore, the effect of the boundary condition is inconsequential. An extension to this study would be to consider boundary conditions that can give rise to transport properties in an Ising model such as, uphill diffusion which can then be compared to the dynamics of the Rayleigh–Bénard system while it is being driven between the end states [47].

## Figures and Tables

**Figure 1 entropy-22-00561-f001:**
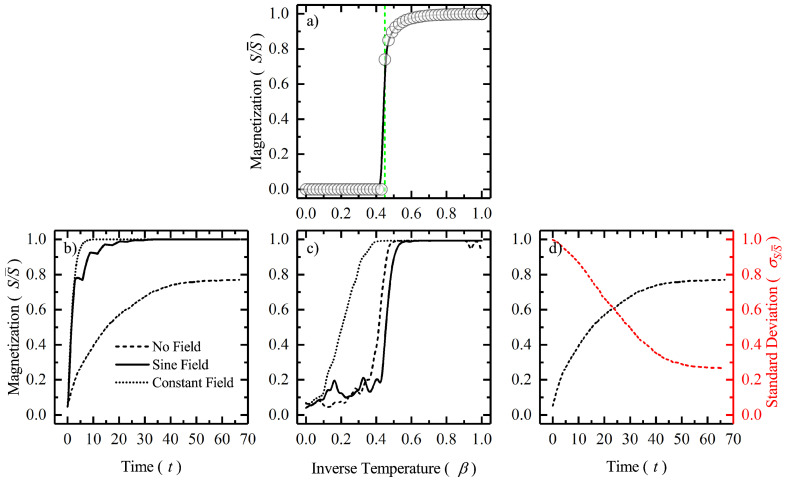
(**a**) Figure shows phase transition in a two-dimensional square lattice Ising model. The magnetization in the system ($S/\bar{S}$) is plotted as a function of the inverse temperature (β). Vertical dotted line denotes $\beta_{c}\approx 0.44$ on the abscissa. (**b**) Figure shows magnetization as a function of time for three cases: h=0 (dashed), h=Asinωt (solid) and h=constant (dotted). (**c**) Figure shows magnetization as a function of inverse temperature for the previous three cases. (**d**) Figure shows magnetization (in black) and standard deviation of magnetization ($\sigma_{S/\bar{S}}$, in red) as a function of simulation time-steps.

**Figure 2 entropy-22-00561-f002:**
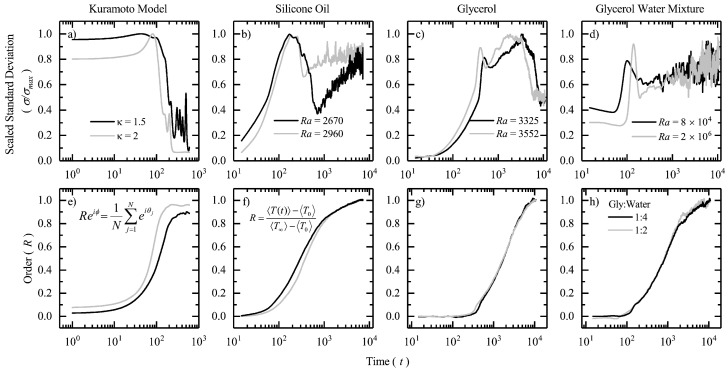
(**a**) Figure shows the scaled standard deviation ($\sigma /\sigma_{max}$) of the angular frequency as a function of time (log-scale) in a two-dimensional Kuramoto system on a lattice for different coupling strengths (κ). (**b**–**d**) Figures show scaled standard deviation of the temperature as a function of time (log-scale) for different fluid samples in a Rayleigh–Bénard convection system. Note that the Rayleigh number (Ra) changes from non-turbulent to turbulent. (**e**–**h**) Figures show the evolution of the order parameter (*R*) as a function of time (log-scale) for the four systems. Note that time is in seconds.

**Figure 3 entropy-22-00561-f003:**
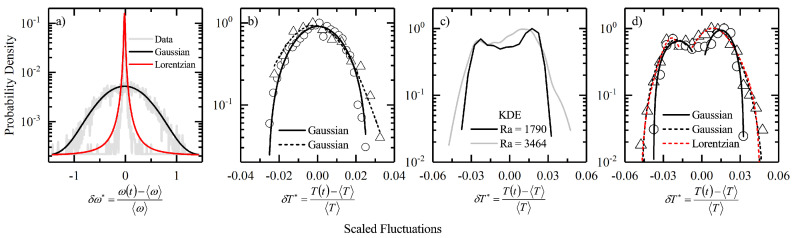
(**a**) Figure shows the probability density functions (log-scale) for the scaled angular frequency fluctuation ($\delta \omega^{⋆}$). The initial randomized state data is fit with a Gaussian (in black) and the final state data is fit with a Lorentzian (in red). (**b**) Figure shows the probability density functions (log-scale) for the scaled thermal fluctuation ($\delta T^{⋆}$) for two different fluid samples at room temperature along with respective Gaussian fits. (**c**) Figure shows the probability density functions (log-scale) for the scaled thermal fluctuation for two different fluid samples at steady-state along with kernel density estimates (KDE). Note that in the final state the two samples correspond to two separate Rayleigh numbers. (**d**) Figure shows the probability density functions (log-scale) for the scaled thermal fluctuation for two different fluid samples at steady-state along with respective Gaussian (in black) and Lorentzian (in red) tails. The absence of sufficient data points prevent us from fitting the final state data of the Ra=1790 sample with a Lorentzian function.

**Figure 4 entropy-22-00561-f004:**
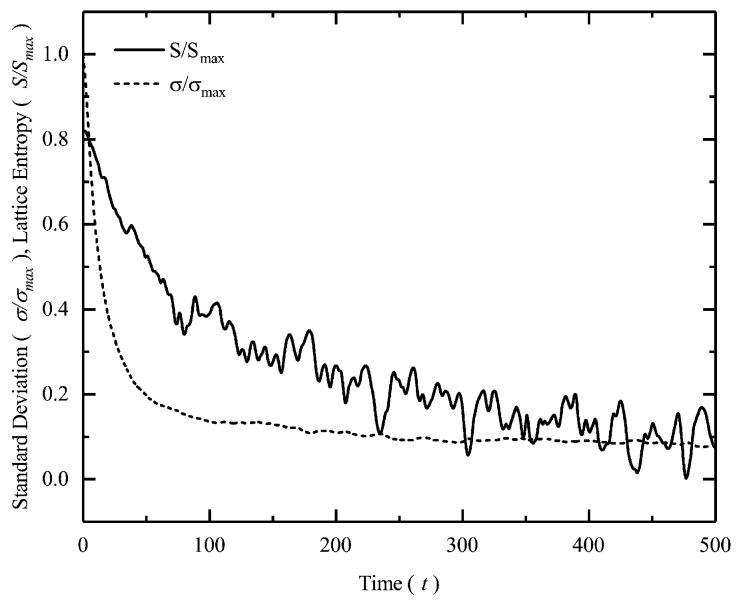
Figure shows scaled standard deviation ($\sigma /\sigma_{max}$) of the angular frequency and lattice entropy ($S/S_{max}$) as a function of time in a two-dimensional Kuramoto system on a lattice.

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
