# Peer review of "An Overview of Emergent Order in Far-from-Equilibrium Driven Systems: From Kuramoto Oscillators to Rayleigh–Bénard Convection"

_entropy, 2020, doi:10.3390/e22050561_

Round 1
Reviewer 1 Report
The authors have addressed most of the concerns I raised in my first review. Still, I believe the paper would benefit from a clear problem definition up front. This is lacking in this version. The authors state that they intend to study the role of "coupling" (Introduction p23) but the nature of coupling is not discussed. Similarly, the discussion of the Ising model (section 1.1) does not give an indication as to what the reader may expect to learn from it in this study. I consider these issues of style rather than substance, which however would improve the readability of the paper by a more general audience.
A few minor typos are listed here but the authors should check the text more carefully:
Lines 30-31: The choice of our systems is not random
Line 33: The coexistence of two states ...
Line 62: Finally, we discuss on of the simplest experimental setups to...
Line 62 (in the middle of the paragraph): one of the most actively and extensively studies physical systems...
Line 62, above Eq. 8: "which is defined by " followed by Eq 8: as written it implies that Eq 8 defines the onset of convection while in reality it defines the Rayleigh number.
Author Response
The authors have addressed most of the concerns I raised in my first review. Still, I believe the paper would benefit from a clear problem definition up front. This is lacking in this version. The authors state that they intend to study the role of "coupling" (Introduction p23) but the nature of coupling is not discussed. Similarly, the discussion of the Ising model (section 1.1) does not give an indication as to what the reader may expect to learn from it in this study. I consider these issues of style rather than substance, which however would improve the readability of the paper by a more general audience.
We have added a few lines to improve the clarity of focus of the work, see abstract and lines 40-45. We have also added a few lines in the text to provide clarity on the role of the Ising model in this paper, see line 53-57. Regarding coupling, we are not sure what exactly the reviewer is expecting but we have clarified in the text that these are examples of ‘mean-field coupling models’. We thank the reviewer in helping us making the paper more accessible to the general audience.
A few minor typos are listed here but the authors should check the text more carefully:
Lines 30-31: The choice of our systems is not random
Line 33: The coexistence of two states ...
Line 62: Finally, we discuss on of the simplest experimental setups to...
Line 62 (in the middle of the paragraph): one of the most actively and extensively studies physical systems...
Line 62, above Eq. 8: "which is defined by " followed by Eq 8: as written it implies that Eq 8 defines the onset of convection while in reality it defines the Rayleigh number.
We have corrected all the above typos and we have thoroughly read through the manuscript for similar errors. We thank the reviewer in pointing them out.
Reviewer 2 Report
The manuscript reviews, in a very coarse grained fashion, three paradigmatic models of nonequilibrium physics and aims at highlighting the few, if any, analogies they share for what concerns the onset of phase transitions.
The manuscript does not give, in my opinion, any relevant insight neither into the physics of the three models (remarkably, the role of the boundary conditions is still completely overlooked here, although I recommended to consider them in my previous report) nor into the mathematical techniques used in the investigation of phase transitions (which are indeed quite peculiar for each model: think, for instance, of the role of Gibbs measures and of the use of the "Peierls argument" in the study of phase transition at low temperatures in the 2D Ising model).
Thus, I find the manuscript useful mainly at a pedagogical level and not suitable for publication on Entropy.
Author Response
The manuscript reviews, in a very coarse grained fashion, three paradigmatic models of nonequilibrium physics and aims at highlighting the few, if any, analogies they share for what concerns the onset of phase transitions.
The manuscript does not give, in my opinion, any relevant insight neither into the physics of the three models (remarkably, the role of the boundary conditions is still completely overlooked here, although I recommended to consider them in my previous report) nor into the mathematical techniques used in the investigation of phase transitions (which are indeed quite peculiar for each model: think, for instance, of the role of Gibbs measures and of the use of the "Peierls argument" in the study of phase transition at low temperatures in the 2D Ising model).
Thus, I find the manuscript useful mainly at a pedagogical level and not suitable for publication on Entropy.
We are not sure about the reviewer’s comments with respect to the revised version of the manuscript. In our previous response, we did clearly state why the role of boundary condition is inconsequential for our steady-state analysis. We have also explained that in text, see lines 252-257. Moreover, the current study is not intended to provide new insights about the physics of these systems that are studied rather the purpose is to show an important connection across these systems. This paper aims to provide a “foundation for modeling a complex system as a two state-system, where the states: order and disorder can coexist as the system is driven away from equilibrium” (see abstract). For independent discussions on the theoretical developments on Ising, Kuramoto and the RBC we have listed numerous references as these have been widely studied for decades.
Reviewer 3 Report
This paper provides some considerations about three models that driven out-of-equilibrium give rise to emergent spatio-temporal order through self-organization.
The paper contains an overview that is well-written and looks suitable for publication after some revision.
I suggest to emphasize, in a more detailed form, the conclusion deriving from the experimental analysis of the models.
Author Response
This paper provides some considerations about three models that driven out-of-equilibrium give rise to emergent spatio-temporal order through self-organization.
The paper contains an overview that is well-written and looks suitable for publication after some revision.
I suggest to emphasize, in a more detailed form, the conclusion deriving from the experimental analysis of the models.
Yes, we have done that see lines 232-239. We thank the reviewer for them comments.